# Design of LTE/Sub-6 GHz Dual-Band Transparent Antenna Using Frame-Structured Metal Mesh Conductive Film

**DOI:** 10.3390/nano13020221

**Published:** 2023-01-04

**Authors:** Yu-Ming Lin, Hung-Wei Wu, Shoou-Jinn Chang

**Affiliations:** 1Department of Photonics, National Cheng Kung University, Tainan 701, Taiwan; 2Department of Electrical Engineering, Feng Chia University, Taichung City 407, Taiwan; 3Micro Nano System Center, School of Information Science and Technology, Fudan University, Shanghai 200433, China; 4Advanced Optoelectronic Technology Center, Department of Electrical Engineering, Institute of Microelectronics, National Cheng Kung University, Tainan 701, Taiwan

**Keywords:** transparent antennas, LTE/Sub-6G, metal mesh, frame structure

## Abstract

This paper proposes a dual-band transparent antenna using frame-structured metal mesh conductive film (MMCF). The frame-structured metal mesh conductive film is based on the conductive-coated thin film and forms a narrow strip surrounding the edge of the antenna. The frame-structured metal mesh conductive film can resist considerable current leakage on the edge of the conductive strip to improve the antenna’s efficiency by 51% at 2.1 GHz and 53% at 3.6 GHz. As a result, the transparent dual-band antenna has an operating bandwidth of 1.9–2.4 GHz and 3.2–4.1 GHz with a high transparency of 80%, which make it valuable to the applications of biomedical electronic components, wearable devices, and automobile vehicles.

## 1. Introduction

With rapid developments in wireless communication technology, the demands for transparent antennas have increased dramatically in applications [1,2]. Some researchers are studying the results of transparency antenna technology and low resistive thin-film materials [3,4,5,6,7,8]. In 2016, Li et al. proposed a dual-band transparent multiple-input and multiple-output antenna using the metal mesh thin film operating on both WLAN bands [3]. In 2017, Kosuga et al. proposed a transparent dipole antenna fabricated on a quartz substrate with monolayer graphene [4]. In 2017, S. Hong et al. presented transparent microstrip patch antennas with multilayer and metal-mesh films. The transparent microstrip patch antennas are made of two types of transparent conductive films: multilayer film (MLF; IZTO/Ag/IZTO) and metal-mesh film (MMF; Cu) [5]. In 2019, Desai et al. presented a dual-band optically transparent antenna based on a slotted interconnected ring resonator as an efficient radiating element that was fabricated on a Silver–Tin–Oxide (AgHT)-based substrate [6]. In 2020, Q. H. Dao et al. presented an optically transparent 24 GHz analog front-end RFID sensor tag. The transparent device is placed on a single efficient solar cell, and the transparency is achieved by realizing metal patches and lines as a grid structure. The tag fabricated on quartz glass has an overall transparency of 75% and an antenna efficiency of 49% [7]. In 2020, R. Yazdani et al. presented a miniaturized triple-band highly transparent antenna. The antenna achieves more than 80% transparency [8]. However, transparent antennas using thin-film technologies could produce current leakages along the edge of the conductive strip, reducing the performance of the transparent antenna is an important issue. In [9], the concept of improving the efficiency of a dual-band transparent antenna using the conductive coating narrow strip on the antenna frame area was proposed. It notes that the etching process of the antenna structure could produce many surface defects on the conductive thin film. These references inspired us to design an antenna with high performance and transparency simultaneously. The proposed dual-band transparent antenna using frame-structured metal mesh conductive film could be effectively applied on mm-wave devices because the current leakage at the metal edge of the device becomes serious when the device operates in the mm-wave frequency range. The frame-structured metal mesh conductive film can reduce the effect of current leakage at the metal edge of the device to ensure a high-performance device in the time-wave range.

This study proposes a dual-band transparent antenna using frame-structured metal mesh conductive thin film to improve the antenna’s performance. The frame-structured metal mesh conductive film (MMCF) is based on the conductive-coated thin film and forms a narrow silver strip along the edge of the antenna. The frame-structured antenna can effectively provide a high current density bounding to improve microwave efficiency and avoid current leakage in the transparent antenna structure. As a result, the dual-band transparent antenna using frame-structured metal mesh conductive thin film is proposed for the first time. The measured results show that the proposed dual-band transparent antenna has more than 51% efficiency at 2.1 GHz and 53% at 3.6 GHz, and it achieves 80% transparency.

## 2. Study of the Frame-Structured Metal Mesh Conductive Film

Figure 1a,b shows a comparison of measured insertion losses (|S_21_|) and calculated conductor losses (α_c_) of the MMCF-based microstrip lines with different widths of the frame structure. The total line attenuation loss (α_t_) of the MMCF-based microstrip line is the sum of the conductor loss (α_c_) and dielectric loss (α_d_) and is determined using [10]:(1)$$\alpha_{t}=8.686\frac{\ln\left(10^{-\frac{\left|S_{21}\right|}{-20}}\right)}{L}=\alpha_{c}+\alpha_{d}\;\left(\mathrm{dB}\cdot {\mathrm{mm}}^{-1}\right)$$
where the conductor loss (α_c_) of the thin film microstrip line is determined using the characteristic impedance Z_0_, the series distributed resistance for the microstrip line R_1_, and the series distributed resistance R_2_ of the ground plane [11] as follows:(2)$$\alpha_{c}=8.686\frac{\left(R_{1}+R_{2}\right)}{2Z_{0}}\;\left(\mathrm{dB}\cdot {\mathrm{mm}}^{-1}\right)$$
where R_1_ and R_2_ are determined using [12]:(3)$$R_{1}=\mathrm{LR}\frac{R_{s}}{w}\left(\frac{1}{\pi }+\frac{1}{\pi^{2}}\ln\frac{4\pi w}{t}\right)\;\left(\Omega \cdot {\mathrm{mm}}^{-1}\right)$$
(4)$$R_{2}=\frac{R_{s}}{w}\left(\frac{\frac{w}{h}}{\frac{w}{h}+5.8+\frac{0.03h}{w}}\right)\;\left(\Omega \cdot {\mathrm{mm}}^{-1}\right)\;,\;0.1<\frac{w}{h}\leq 10$$
where the loss ratio LR is calculated using:(5)$$\mathrm{LR}=0.94+0.132\frac{w}{h}-0.0062{\left(\frac{w}{h}\right)}^{2},\;0.5<\frac{w}{h}\leq 10$$
where w/h is the line width/substrate thickness, t is the thickness of the conductor, and the skin effect resistance R_s_ = 1/δσ (δ is the skin depth and σ is the conductivities of the conductive thin film). The conductor loss influenced by the skin effect at different conductive material thicknesses and geometries depends on R_1_, and the conductor loss affected by the ground plane and geometry depends on R_2_. In Figure 1a, the MMCF-based microstrip line with a 0.5 mm width of the frame structure has a similar insertion loss to the copper-based microstrip line; therefore, the frame structure can effectively reduce the current leakage on the edge of the thin film transmission lines. The low conductor loss of the MMCF-based microstrip lines with different frame structure types is shown in Figure 1b. Figure 1c shows the surface current distributions of the MMCF-based microstrip line. The current distribution of the copper-based microstrip line has an extreme current concentration bounded in the conductive line. On the other hand, when using an MMCF-based microstrip line with a 0.5 mm width of the frame structure, the current distribution of the conductive line shows the current solid concentration at the operating frequency. Therefore, the high-performance transparent thin-film antenna can be realized by using the proposed frame structure.

Figure 2 shows the top view and 5× objective observation using the NIKON optical microscope platform. A conductive low-temperature silver glue with a sheet resistance of less than 0.02 Ω/sq was used for the 50 Ω SMA connectors to the antenna feedlines. An Agilent N5230A vector network analyzer was used to measure the dual-band transparent antenna. The antenna was simulated and designed using the Ansys HFSS EM simulation.

Figure 3a,b shows the top view of the proposed dual-band transparent antenna with frame-structured metal mesh conductive film and a photograph of the fabricated antenna. A 0.5 mm thick Corning glass substrate with a relative permittivity of 5.27 and a loss tangent of 0.003 was used. This study used a transparent thin film as the metal mesh conductive film (MMCF) [13]. The sheet resistance of the metal mesh conductive film is 0.05 Ω/sq, which is much lower than 10 Ω/sq of indium tin oxide (ITO), carbon nanotube (CNT), and AgHT-series materials. The MMCF thin film has a conductive layer and a polyethylene terephthalate layer.

The thick film deposition process forms the frame-structured metal mesh conductive film. In this study, the frame-structured, 50 μm-thick, was coated using the MMCF/corning glass substrate coating blade, where the frame-structured coating speed and the substrate temperature were kept at 20 mm/s and 25 °C, respectively. A mask was patterned using 60 μm-thick heat-resistant tape. The dual-band transparent antenna has dual monopoles on the top layer of the substrate and the ground plane with an area of 20 × 10 mm^2^ on the bottom layer of the substrate, as shown in Figure 1. The design of each monopole has a length of 29 mm (0.25 λ_g_ at 2.1 GHz and 0.75 λ_g_ at 3.6 GHz, where λ_g_ is the guided wavelength at the operating frequency) to generate the dual-band frequency response. The circuit size of the dual-band transparent antenna is 50 × 50 mm^2^. The dual monopoles are designed to be separated by 14 mm to avoid unwanted cross-coupling effects.

## 3. Simulation and Measured Results

Figure 4 shows a comparison of the current distributions of the dual-band transparent antennas with and without using the proposed frame structure at 2.1/3.6 GHz. The surface current concentration of the dual-band transparent antenna with frame structure is stronger than the antenna without using the frame structure. The use of a frame-structured antenna can effectively avoid the current leakage along the edge of the conductive strip of the antenna; therefore, both the high transmittance and high efficiency of the antenna can be achieved simultaneously.

Figure 5 shows the measured and simulated S-parameters of the dual-band transparent antenna. Both port 1/port 2 are terminated with a 50-ohm (Ω) matching load. The measured results of return losses (−20 log |S_11_|) of the antenna at 2.1 and 3.6 GHz were found at 25.6 dB and 15.2 dB, respectively. The measured |S_11_| are all below 10 dB at 1.9 to 2.4 GHz and 3.4 to 4.1 GHz, which were in agreement with the simulated results and enough for good performance in 5G applications [14]. The measured bandwidths of the lower frequency bands were 2.36–2.54 GHz and 2.35–2.53 GHz, respectively. The discrepancy in the 2D radiation pattern between the simulated and measured results is mainly due to two factors: (1) the cable and prototype assembly lead to differences; (2) the SATIMO microwave anechoic chamber measurement equipment, StarLab, is difficult to maintain at precisely the same conditions as the EM simulation model. Figure 6 shows the simulated frequency shift capability with different lengths L_a_ and L_b_ from 1 to 5 mm and 9 to 13 mm, respectively.

Figure 7 shows the measured envelope correlation coefficient (ECC, ρ_e_) and microwave radiation efficiency of the dual-band transparent antenna with and without the frame structure. We proposed the dual-band transparent antenna is in working operation and calculated its corresponding ECC to evaluate its performance [15]. The ECC is used to describe how many communication channels are isolated or correlated with each other in a system [16]. The envelope correlation coefficient of the antenna can be found using the equations as described below:
(6)$$\rho_{e}=\frac{{\left|S_{11}^{*}S_{12}+S_{21}^{*}S_{12}\right|}^{2}}{\left(1-{\left|S_{11}\right|}^{2}-{\left|S_{21}\right|}^{2}\right)\;\left(1-{\left|S_{22}\right|}^{2}-{\left|S_{12}\right|}^{2}\right)}$$
(7)$$\rho_{e}=\frac{{\left|\iint_{4\pi }^{\;}\left[\vec{F_{1}}\left(\theta ,\varphi \right)\cdot \vec{F_{2}}\left(\theta ,\varphi \right)\right]d\Omega \right|}^{2}}{\iint_{4\pi }^{\;}{\left|\vec{F_{1}}\left(\theta ,\varphi \right)\right|}^{2}d\Omega \;\iint_{4\pi }^{\;}{\left|\vec{F_{2}}\left(\theta ,\varphi \right)\right|}^{2}d\Omega }$$

The ECC values for 2.1 and 3.6 GHz are lower than 0.5. To achieve a better performance, the envelope correlation coefficient of the antenna should have a value < 0.5 so that high performance and diversity for mobile terminal applications can be achieved [17]. The measured radiation efficiency of the antenna is shown in Figure 7. The measured efficiency is the average of the measured results of the dual-band transparent antenna using the frame structure with a high radiation efficiency of 51/53 at 2.1/3.6 GHz. The proposed frame structure applied to the antenna can enhance the current concentration on the conductive stripline at operating frequencies and avoid current leakages along the edge of the metal mesh conductive film. Therefore, the microwave performance of the antenna can be further improved. It can be seen that the measured radiation efficiencies of using the frame structure are better than those without using the frame structure. In addition, the proposed antenna’s ground plane uses the frame structure to ensure that the current distribution is uniform without leakage on the edge of the ground plane, as shown in Figure 4.

In this paper, the diversity gain (DG) is another import parameter which assures good diversity and MIMO performance. DG is calculated using Equations (8) and (9) [17,18], and the results are shown in Figure 8.
(8)$$\mathrm{DG}=10e_{\rho }$$
(9)$$e_{\rho }=\sqrt{\left(1-{\left|0.99e_{\rho }\right|}^{2}\right)}$$

In this paper, the mean effective gain (MEG) followed that found in [19]. A solution to this problem was proposed in [20], where a probabilistic environmental model was proposed. Using the environmental model of three-dimensional radiation patterns and the proposed statistical model, one can numerically obtain the mean effective gain by solving a mathematical expression that combines the two quantities. This numerical method allows us to calculate the mean effective gain using the simulated/measured gain patterns in an ideal environment (i.e., the simulation tool or an anechoic chamber), and a model of the environment suitable for the application for which the antenna is being designed. The mathematical expressions for the mean practical gain calculation are shown in Equations (10) and (11):(10)$$\mathrm{MEG}=\overset{2\pi }{\underset{0}{\int }}\overset{\pi }{\underset{0}{\int }}\left[\frac{\mathrm{XPD}}{1+\mathrm{XPD}}G_{\theta }\left(\theta ,\varphi \right)P_{\varphi }\left(\theta ,\varphi \right)+\frac{1}{1+\mathrm{XRD}}G_{\theta }\left(\theta ,\varphi \right)P_{\varphi }\left(\theta ,\varphi \right)\right]\sin\theta d\theta d\varphi$$

The following conditions must be satisfied:(11)$$\int_{0}^{2\pi }\int_{0}^{2\pi }\left[G_{\theta }\left(\theta ,\varphi \right)+G_{\theta }\left(\theta ,\varphi \right)\right]\sin\theta d\theta d\varphi =4\pi \;;\;\int_{0}^{2\pi }\int_{0}^{2\pi }P_{\theta }\left(\theta ,\varphi \right)\sin\theta d\theta d\varphi =\int_{0}^{2\pi }\int_{0}^{2\pi }P_{\theta }\left(\theta ,\varphi \right)\sin\theta d\theta d\varphi =1\;$$
(12)$$\mathrm{XPD}=\frac{P_{V}}{P_{H}}$$
where XPD is the cross-polarization power ratio (or cross-polarization discrimination), which represents the distribution of the incoming power (the ratio between the vertical mean incident power to the horizontal mean incident power), G_θ_ (θ,φ) and G_φ_ (θ,φ) are antenna gain components, and P_θ_ (θ,φ) and P_φ_ (θ,φ) represent the statistical distribution of the incoming waves in the environment, assuming that the two are not correlated. The equations in Equation (11) represent the conditions needed to evaluate Equation (10). A more general formulation using the polarization matrix and other incoming wave distributions can be found in [19]. Table 1 presents a comparison of the proposed single antenna with others reported in the literature. This comparison table summarized the comparison elements of MIMO with those reported in the literature [3,4,5,6,20,21,22].

Figure 9 shows the measured and simulated radiation pattern of the dual-band transparent antenna with and without the frame structure. The measured radiation pattern of the dual-band transparent antenna has an average gain of −9.5 dB at 2.1 GHz and −6.1 dB at 3.6 GHz. For the measured and simulated radiation patterns of the dual-band transparent antenna at 2.1 GHz and 3.6, good agreement between the measured results can be observed. The antenna has omnidirectional radiation patterns in the E-planes and H-planes for the dual-band frequency response at both 2.1 and 3.6 GHz. The comparison of the proposed transparent antenna with previous works is summarized in Table 1. This study provides a simple fabrication process, high transparency of 80%, and good microwave radiation efficiency of 51% and 53% at 2.4 GHz and 3.6 GHz, respectively, using the frame-structured metal mesh conductive film.

## 4. Conclusions

A dual-band transparent antenna with frame-structured metal mesh conductive film has been successfully presented. The frame-structured metal mesh conductive film is fabricated on the conductive-coated thin film and forms a narrow strip surrounding the edge of the antenna. The proposed frame-structured metal mesh conductive film can resist considerable current leakage on the edge of the antenna’s conductive strip to improve radiation efficiency. The measured results of the dual-band transparent antenna with high efficiency of 51%/53% at 2.1/3.6 GHz, respectively, and obtained transparency of 80% are proposed. The antenna has potential for biomedical electronic components, wearable devices, and automobile vehicles.

## Figures and Tables

**Figure 1 nanomaterials-13-00221-f001:**
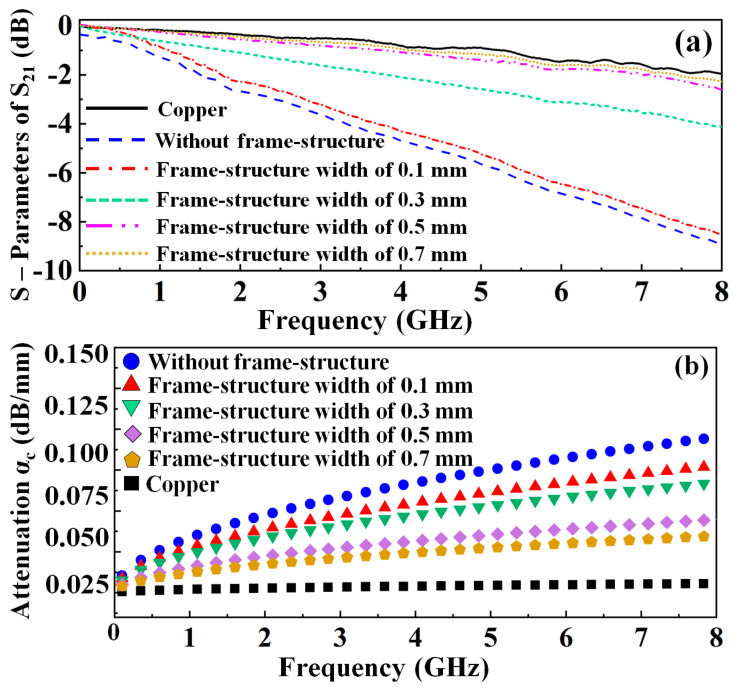
Comparison of the (**a**) measured |S_21_| and (**b**) calculated conductor losses (α_c_) (**c**) transparency conditions of glass, PET, MMCF, and (**d**) surface current distributions of MMCF−based microstrip lines using a different frame structure.

**Figure 2 nanomaterials-13-00221-f002:**
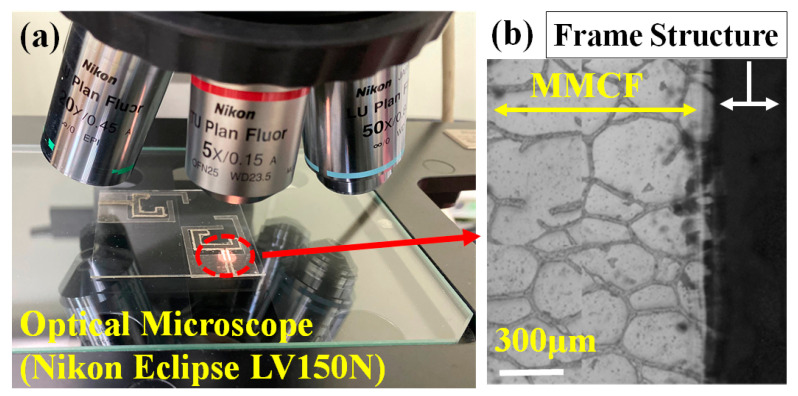
(**a**) Top view and (**b**) 5× objective observation using the NIKON optical microscope platform.

**Figure 3 nanomaterials-13-00221-f003:**
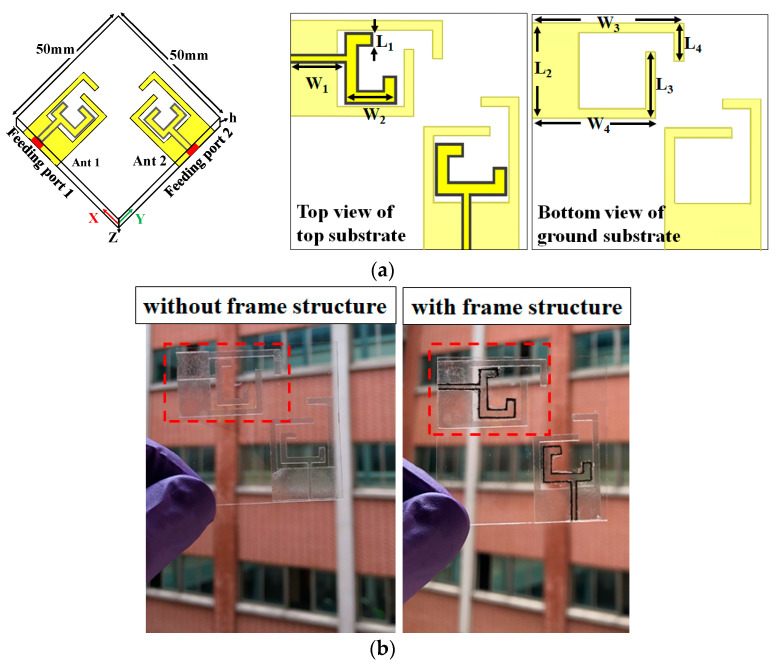
(**a**) Configuration of the proposed dual-band transparent antenna using the frame-structured metal mesh conductive film and (**b**) photograph of the fabricated antenna. (L_1_ = 3, L_2_ = 20, L_3_ = 14, L_4_ = 8, W_1_ = 11.5, W_2_ = 11.5, W_3_ = 32, W_4_ = 26, h = 0.5, W_f_ = 0.5, h_c_ = 0.003, all are in mm).

**Figure 4 nanomaterials-13-00221-f004:**
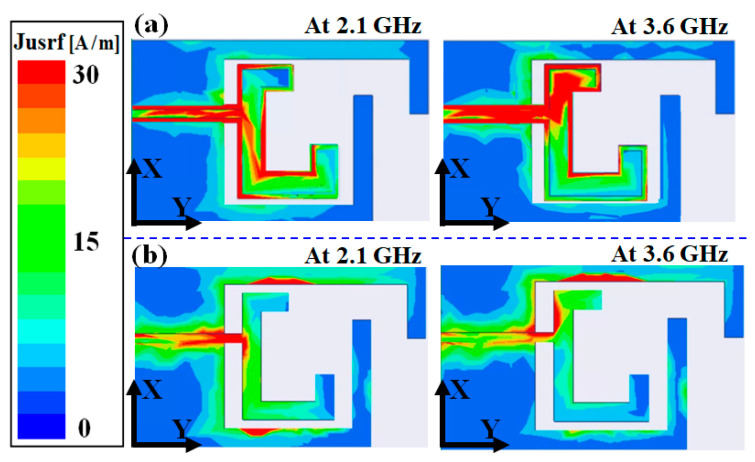
Surface current distributions of the dual-band transparent antennas (**a**) with the frame structure and (**b**) without the frame structure.

**Figure 5 nanomaterials-13-00221-f005:**
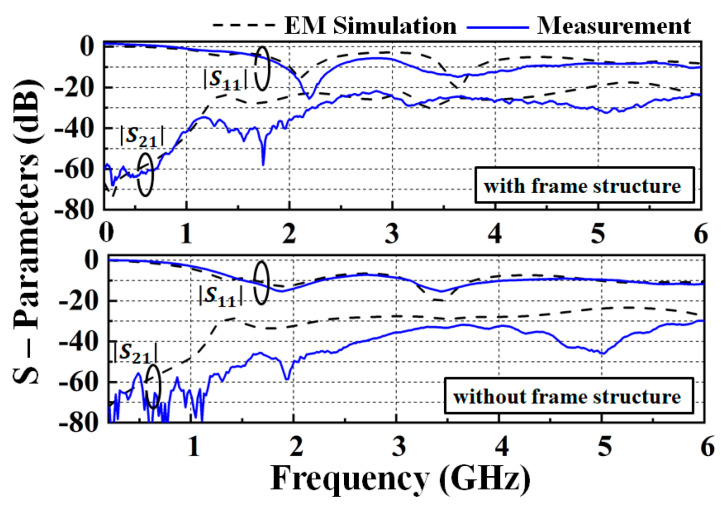
Measured (solid line) and simulated (dash line) S−parameters of the dual−band transparent antenna.

**Figure 6 nanomaterials-13-00221-f006:**
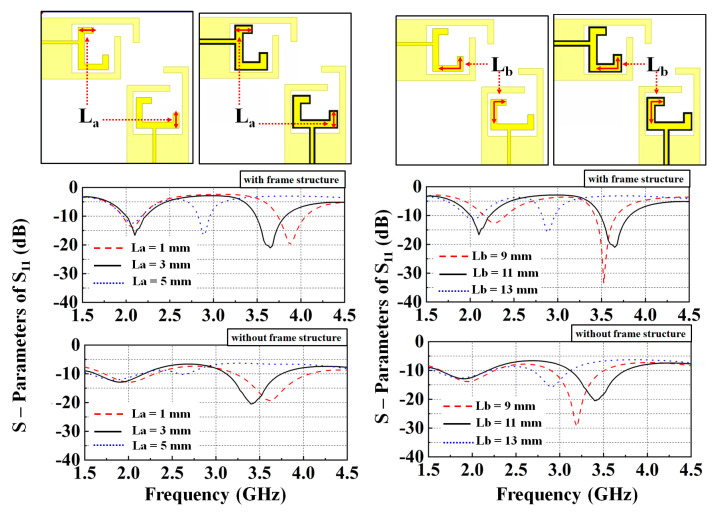
Simulated frequency shift capability with different lengths L_a_ and L_b_ from 1–5 mm and 9–13 mm, respectively.

**Figure 7 nanomaterials-13-00221-f007:**
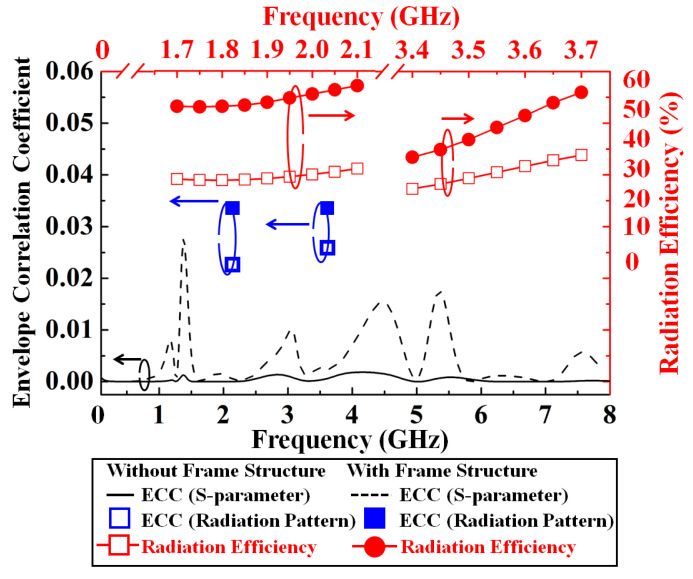
Measured envelope correlation coefficient (ECC) and microwave radiation efficiency of the dual-band transparent antenna with and without using the frame structure.

**Figure 8 nanomaterials-13-00221-f008:**
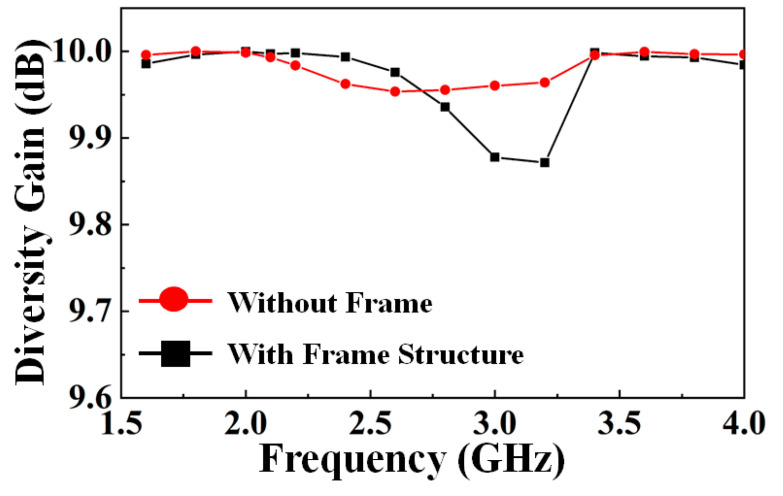
Diversity gain with and without frame structure.

**Figure 9 nanomaterials-13-00221-f009:**
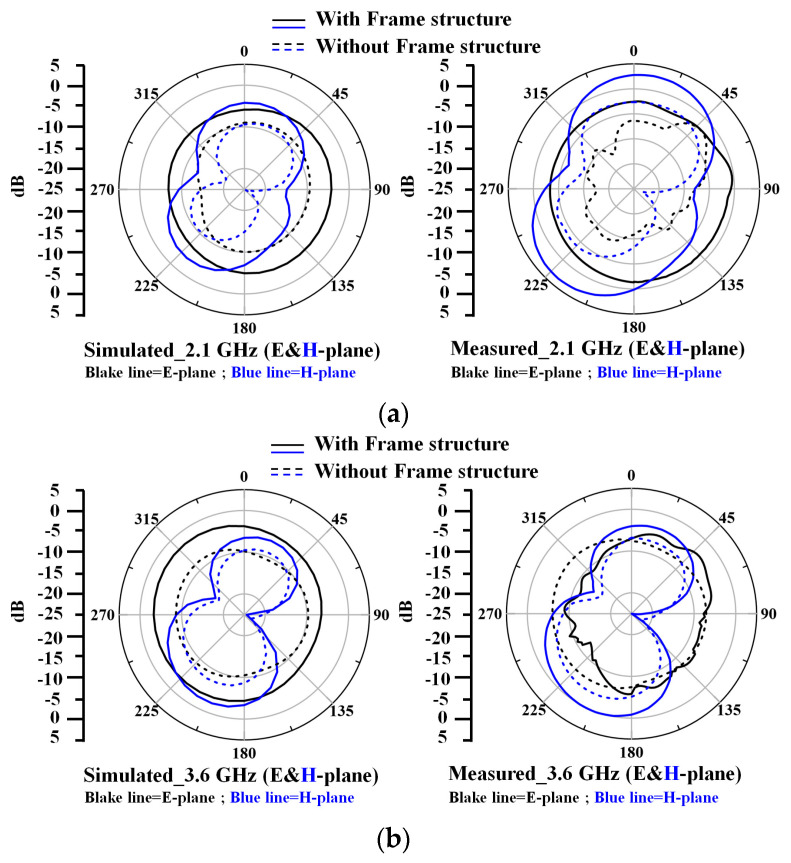
Measured and simulated 2D radiation pattern of the dual-band transparent antenna (**a**) at 2.1 GHz and (**b**) at 3.6 GHz using the frame structure (solid line) and without using the frame structure (dash line).

**Table 1 nanomaterials-13-00221-t001:** Performance comparison of this study with other previous transparent antenna references.

Ref.	Film Structure/Substrate	f_0_ (GHz)	|S_11_| (dB)	Efficiency (%)	Gain (dB)	Transmittance (%)
[3]	MMCF/Glass	2.44/5.5	25/18	43/46	0.74/2.3	75
[4]	Graphene and Au/Glass	20.7	11	×	×	×
[5]	MMF/Acrylic	2.47	15	42.7	2.63	61.5
[6]	Metal Mesh/Glass	24	19	49	4.4	75
[20]	Copper/PDMS	2.43/5.15	27.8/18	37/44	2.2/3.02	70
[21]	AgITO/Glass	3.5/5.8	27/23	×	−2.9/−3	52
[22]	Copper/FR-4	3.54/6.33	32/23	75.1/74.2	3.25/3.4	0
This work	frame-structured MMCF/Glass	2.1/3.6	25.6/15.2	51/53	−9.5/−6.1	80

MMCF: Metal Mesh Conductive Film; PDMS: Polydimethylsiloxane; MMF: Metal Mesh Film; Acrylic: Poly(methyl methacrylate); FR-4: Glass-reinforced Epoxy Laminate material.

## Data Availability

Data are contained within the article.

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
