# Peer review of "Design of LTE/Sub-6 GHz Dual-Band Transparent Antenna Using Frame-Structured Metal Mesh Conductive Film"

_nanomaterials, 2023, doi:10.3390/nano13020221_

Round 1

Reviewer 1 Report

Nice job on this paper.  It is a very good topic and applicable to many applications that call for transparent antennas.  The organization and quality of your paper is very good.  The grammar can be improved with a thorough review.  The only technical comment i have is that you showed the ECC results based on S-parameter measurements.  Why didn't you also compute them based on the antenna elements radiation pattern?  I believe these results would improve the paper.

Author Response

Comments and Suggestions for Authors:

Nice job on this paper.  It is a very good topic and applicable to many applications that call for transparent antennas.  The organization and quality of your paper is very good.  The grammar can be improved with a thorough review.  The only technical comment i have is that you showed the ECC results based on S-parameter measurements.  Why didn't you also compute them based on the antenna elements radiation pattern?  I believe these results would improve the paper.

Author response: Thank you for the helpful comment. We have carefully amended the revision by using colored markings and underlines. ECC using far-field pattern Equation (7) is found to be 0.02357, 0.03258, 0.02664, 0.03358 at 2.1 and 3.6 GHz, respectively, and is shown as bull square in Figure 6. This figure also shows that ECC values are well below 0.5 in the desired frequency bands, which ensures good diversity performance [15]. 

Reviewer 2 Report

The article presents design of LTE/Sub-6 GHz dual-band transparent antenna using frame-structured metal mesh conductive film. Work is interesting and following comment will be helpful to improve the paper.

Introduction section is too short and need to be expanded with more literature insight and by highlighting the short comings and areas that needs to be improved.

Clearly highlight the contributions and key findings to demonstrate the novelty.

It will be great to expand the results section with more design parametric analysis in addition to frame width.

Add results for other MIMO performance parameters i.e., Channel Capacity Loss (CCL), Diversity Gain (DG) and Mean Effective Gain (MEG)

Table 1 present comparison of proposed single antenna with others reported in literature. Add a comparison table to compare 2 elements MIMO with reported in literature.

Author Response

Comments and Suggestions for Authors:

The article presents design of LTE/Sub-6 GHz dual-band transparent antenna using frame-structured metal mesh conductive film. Work is interesting and following comment will be helpful to improve the paper. Introduction section is too short and need to be expanded with more literature insight and by highlighting the short comings and areas that needs to be improved. Clearly highlight the contributions and key findings to demonstrate the novelty. It will be great to expand the results section with more design parametric analysis in addition to frame width. Add results for other MIMO performance parameters i.e., Channel Capacity Loss (CCL), Diversity Gain (DG) and Mean Effective Gain (MEG)

Author response: Thank you for the helpful comment. We have carefully amended the revision by using colored markings and underlines. We have added three references into the introduction section to enhance our research contribution and importance. This study proposes a dual-band transparent antenna using frame-structured metal mesh conductive thin film to improve the antenna's performance. The frame-structured metal mesh conductive film (MMCF) is based on the conductive-coated thin film and forms a narrow silver strip along the edge of the antenna. The frame-structured antenna can effectively provide a high current density bounding in the antenna to improve the microwave efficiency and avoid current leakage in the transparent antenna structure. As a result, the dual-band transparent Antenna using frame-structured metal mesh conductive thin film is proposed for the first time. The measured results show that the proposed du-al-band transparent antenna has more than 51% efficiency at 2.1 GHz and 53% at 3.6 GHz, and achieves 80% transparency, respectively. The proposed dual-band transparent antenna using frame-structured metal mesh con-ductive film could be effectively applied on the mm-wave devices because the current leakage at the metal edge of the device becomes serious when the device operates on the mm-wave frequency range. The frame-structured metal mesh conductive film can reduce the effect of current leakage at the metal edge of the device to ensure a high-performance device at the time-wave range.

In Figure 1, we have added the conditions of frame-structure width from 0.1 to 0.7 mm for observing the conductor losses changes. Also, the transmittance of different substrate materials is shown in Figure 1(c).

Figure 1. Comparison (a) measured |S21| and (b) calculated conductor losses (αc), (c) transparency conditions of glass, PET, MMCF, and (d) surface current distributions of MMCF-based microstrip lines using different frame structure.

Figure 6. Simulated frequency shift capability with different length La and Lb from 1 to 5 mm and 9 to 13mm, respectively.

In this paper, the diversity gain (DG) is another import parameter which assures good diversity and MIMO performance. DG is calculated using equation (8,9) [17-18].

 Figure 8. Diversity gain of with and without frame structure.

In this paper, the mean effective gain (MEG) can be founded in [19]. A solution to this problem was proposed in [20], where a probabilistic environmental model was proposed. Using the environmental model of three-dimensional radiation patterns and the proposed statistical model, one can numerically get mean effective gain by solving a mathematical expression that combines the two quantities. This numerical method allows us to get mean effective gain using the simulated/measured gain patterns in an ideal environment (i.e., the simulation tool or an anechoic chamber), and a model of the environment suitable for the application for which the antenna is being designed. The mathematical expressions for the mean practical gain calculation are shown in equation (10) and (11):

The following conditions must be satisfied:

 where XPD is the cross-polarization power ratio (or cross-polarization discrimination), which represents the distribution of the incoming power (the ratio between the vertical mean incident power to the horizontal mean incident power), Gθ (θ ,φ ) and Gφ (θ ,φ ) are antenna gain components, and Pθ (θ ,φ ) and Pφ (θ ,φ ) represent the statistical distribution of the incoming waves in the environment, assuming that the two are not correlated. The equations in Equation (11) represent the conditions needed for evaluating Equation (10). A more general formulation using the polarization matrix and other incoming wave distributions can be found in [19]. Table 1 present comparison of proposed single antenna with others reported in literature. Add a comparison table to compare 2 elements MIMO with reported in literature.

Table 1. Performance comparison of other previous Transparent antenna references.

Ref.

Film Structure / Substrate

f0 (GHz)

|S11| (dB)

Efficiency (%)

Gain (dB)

Transmittance (%)

[3]

MMCF / Glass

2.44 / 5.5

25 / 18

43 / 46

0.74 / 2.3

75

[4]

Graphene and Au / Glass

20.7

11

×

×

×

[5]

MMF / Acrylic

2.47

15

42.7

2.63

61.5

[6]

Metal Mesh / Glass

24

19

49

4.4

75

[21]

Copper / PDMS

2.43 / 5.15

27.8 / 18

37 / 44

2.2 / 3.02

70

[22]

AgITO / Glass

3.5 / 5.8

27 / 23

×

-2.9 / -3

52

[23]

Copper / FR-4

3.54 / 6.33

32 / 23

75.1 / 74.2

3.25 / 3.4

0

This work

frame-structured MMCF/ Glass

2.1 / 3.6

25.6 / 15.2

51 / 53

-9.5 / -6.1

80

MMCF : Metal Mesh Conductive Film ; PDMS : Polydimethylsiloxane ; MMF : Metal Mesh Film ; Acrylic : Poly(methyl methacrylate) ; FR-4 : Glass-reinforced Epoxy Laminate material

Round 2

Reviewer 2 Report

Authors have addressed reviewer's comments and manuscript has been improved.